# Five Post-Translational Modification Residues of CmPT2 Play Key Roles in Yeast and Rice

**DOI:** 10.3390/ijms24032025

**Published:** 2023-01-19

**Authors:** Jiayi Tang, Chen Liu, Yiqing Tan, Jiafu Jiang, Fadi Chen, Guosheng Xiong, Sumei Chen

**Affiliations:** 1College of Horticulture, Nanjing Agricultural University, Nanjing 210095, China; 2Academy for Advanced Interdisciplinary Studies, Nanjing Agricultural University, Nanjing 210095, China; 3State Key Laboratory of Crop Genetics and Germplasm Enhancement, Key Laboratory of Flower Biology and Germplasm Innovation, Ministry of Agriculture and Rural Affairs, Key Laboratory of Biology of Ornamental Plants in East China, National Forestry and Grassland Administration, College of Horticulture, Nanjing Agricultural University, Nanjing 210095, China; 4Nanjing Institute of Agricultural Sciences, Jiangsu Academy of Agricultural Sciences, Nanjing 210046, China; 5College of Life Science, Nanjing Agricultural University, Nanjing 210095, China

**Keywords:** *CmPT2*, yeast complementation, post-translational modification, rice

## Abstract

Chrysanthemum (*Chrysanthemum morifolium* Ramat.) is one of the largest cut flowers in the world. Phosphate transporter Pht1 family member CmPht1;2 protein (CmPT2) plays an important role in response to low-phosphate (LP) stress in chrysanthemum. Post-translational modification (PTM) can modulate the function of proteins in multiple ways. Here, we used yeast and rice systems to study the role of putative PTM in CmPT2 by determining the effect of mutation of key amino acid residues of putative glycosylation, phosphorylation, and myristoylation sites. We chose nine amino acid residues in the putative PTM sites and mutated them to alanine (A) (*Cmphts*). CmPT2 recovered the growth of yeast strain MB192 under LP conditions. However, G84A, G222A, T239A, Y242A, and N422A mutants could not grow normally under LP conditions. Analysis of phosphorus absorption kinetics showed that the Km of CmPT2 was 65.7 μM. Among the nine *Cmphts*, the expression of five with larger Km (124.4–397.5 μM) than CmPT2 was further evaluated in rice. Overexpression of *CmPT2*-OE increased plant height, effective panicle numbers, branch numbers, and yield compared with that of wild type ‘Wuyunjing No. 7’ (W7). Overexpression of *Cmphts*-OE led to decreased plant height and effective panicle numbers compared with that of the *CmPT2*-OE strain. The Pi content in roots of *CmPT2*-OE was higher than that of the W7 under both high (normal) phosphate (HP) and LP conditions. However, the Pi content in the leaves and roots was significantly lower in the N422A-OE strain than in the *CmPT2*-OE strain under both HP and LP conditions. Under LP conditions, the phosphorus starvation response (PSR) genes in *CmPT2*-OE were inhibited at the transcription level. The expression patterns of phosphorus-related genes in T239A, Y242A, and N422A-OE under LP conditions were different from those of *CmPT2*-OE. In conclusion, these five post-translational modification residues of CmPT2 play key roles in modulating the function of CmPT2. This work boosters our understanding of the function of phosphate transporters and provides genetic resources for improving the efficiency of phosphorus utilization in crop plants.

## 1. Introduction

Phosphorus (P) is one of the three essential mineral nutrients required for plants [1]. It is involved in the vast majority of plant metabolic pathways, including energy metabolism, nucleic acid synthesis, signal transduction, and phosphorylation [2]. Soluble phosphates (HPO_4_^2−^ or H_2_PO_4_^−^) are the main forms of phosphorus taken up and utilized by plants. The immobilization of most phosphate from fertilizer in the soil and the organic forms converted by soil bacteria increase the difficulty of inorganic phosphate (P_i_) absorption by plants [3]. Crucially, the efficiency of phosphate uptake from the soil to plants is a limiting factor hindering crop yields. To compensate for the low efficiency of phosphate absorption by plants, excess phosphorus fertilizer is applied to farmland [4]. This not only increases production costs but also leads to a series of problems, such as land hardening, crop quality decline, and water eutrophication [5].

The phosphate transporter 1 (Pht1) family is the most important membrane-localized phosphate transporter for phosphate absorption and transport in plants [6]. Research on phosphorus absorption and transport mechanisms has increased, both in crops, such as rice, and model plants [7]. Post-translational modifications (PTMs) increase the functional diversity of proteins through the covalent addition of functional groups or proteins, proteolytic cleavage of regulatory subunits, or degradation of the entire protein [8]. Pht1 is also subject to a variety of post-translational regulatory processes, such as myristoylation, glycosylation, and phosphorylation [9]. Myristoylation (MYR) of glycine (G) is ubiquitous in eukaryotes, associated with different membrane compartments, and also likely to influence cellular homeostasis [10]. Glycosylation of asparagine (N) suppresses inducible nitric oxide synthase activity by disturbing electron transfer in mice and is important for the slow in vivo clearance of recombinant human diamine oxidase heparin-binding motif mutants [11]. The phosphorylation of serine (S), tyrosine (Y), and threonine (T) is an integral component of signal transduction pathways in eukaryotic cells [12]. While a handful of studies have focused on the function of these PTMs in plants Pht1 family, many questions remain, demanding further research be carried out.

Chrysanthemum (*Chrysanthemum morifolium* Ramat.) is an important species of cut flower in autumn with rich ornamental, medicinal, and dietary value. Phosphorus deficiency reduces the development of lateral branches and buds of chrysanthemum, thereby reducing chrysanthemum yield. CmPht1;2 (CmPT2, OQ282146) is a Pht1 family Pi transporter in chrysanthemum. *CmPT2* from the chrysanthemum cultivar ‘Nannongyinshan’ overexpression lines could effectively improve the growth of the low phosphate (LP, 10 μM Pi)-sensitive chrysanthemum cultivar ‘Jinba’ under LP conditions [13]. The CmPT2 protein contains glycosylation, phosphorylation, and myristoylation acylation amino acid PTM sites. The effect of these potentially modified sites of CmPT2 on phosphate transport function is yet to be elucidated. In the present study, we created nine PTM mutants (*Cmphts*) of CmPT2 using the point mutation method. However, since chrysanthemum is polyploid, it is difficult to directly compare the difference in phosphate transport function of *Cmphts* with CmPT2 due to the influence of the endogenous CmPT2 protein. To address this challenge, we used the yeast PHO84 deletion mutant MB192. *CmPT2* and *Cmphts* were introduced into yeast mutant strains to verify the phosphate transport function of PTMs in CmPT2. In addition, we filtered the key PTMs that did not complement the growth of MB192 and transferred them into the rice cultivar ‘Wuyunjing No. 7’ (W7). Our study may help explore key amino-acid modification sites and elucidate the function of PTMs in CmPT2 related to phosphate transport.

## 2. Results

### 2.1. Site-Directed Mutagenesis and the Efficacy of Mutated Forms of CmPT2

According to the structure of the membrane-spanning domain and analysis of CmPT2 PTM sites (myristoylation, glycosylation, phosphorylation), we mutated nine key amino acid sites, including G, S, T, and Y, and N, into A, a nonpolar amino acid. The locations and sequences of the nine site-directed mutants are shown in Figure 1.

### 2.2. Yeast Complementation

The verification of complementary growth indicated that yeast strain MB192 cells heterologously expressing CmPT2 (Yp112-*CmPT2*) and WT grew well under both 20 and 60 μM Pi (LP) conditions. G84A, G222A, T239A, Y242A, and N422A of the *Cmphts* yeast transformant did not grow well under LP conditions and the transformation of empty carrier p112A1NE, whereas others had no obvious change with Yp112-*CmPT2*. However, under high phosphorus (HP, 100 μM Pi and YPDA) conditions, all yeast transformants grew normally (Figure 2). According to the OD_600_ value curves of the WT, Yp112, Yp112-*CmPT2*, and Yp112-*Cmphts* at different culture times, the growth rate of Yp112-*CmPT2* was significantly higher than that of Yp112. However, it did not reach the growth rate of the wild type, indicating that CmPT2 can partially restore the growth of mutant MB192. Cultures with point mutations G84A, G222A, T239A, Y242A, and N422A did not grow as well as Yp112-*CmPT2* at different culture times under LP conditions (60 μM) (Figure 3). Yp112-*CmPT2*, WT, and most point mutation yeast strains grew best at a pH of 4.0. However, T239A mutants were little affected by changes in pH. N422A mutants grew best at pH 7.0. G449A and G456A transformants grew best at pH 6.0. The S510A transformant grew best at a pH of 5.0 (Figure 4). The dynamic curves of the point mutation transformants G84A, G222A, T239A, Y242A, and N422A all deviated from normal Michaelis–Menten kinetics. Radioisotope ^32^P-labeled phosphate determination of K_m_ of Yp112-*CmPT2* was 65.7 µM, with a maximum absorption rate of 170 pmol P_i_ (mg yeast cells min^−1^). In the mutants, the kinetic curves of G84A, G222A, T239A, Y242A, and N422A did not conform to the Michaelis–Menten equation. The K_m_ values ranged from 124.4 to 397.5 µM, much larger than the K_m_ value of Yp112-*CmPT2* (Figure 5).

### 2.3. PCR and RT-PCR Identification of the Transgene Rice

We transformed the *CmPT2* and its five point mutants that did not complement the growth of MB192 into wild type W7. DNA was extracted from the T1 and T2 generations of hygromycin-resistant plants using sodium dodecyl sulfate (SDS), and the transgenic plants were identified with primers *CmPT2*-1100-F and GFP-R with a target fragment size of approximately 590 bp. The results are presented in Appendix A. Among the 17 *CmPT2*-OE lines, OE-8-4, OE-11-1, OE-11-5, and OE-11-11 were all homozygous, while the others were heterozygous. Three G84A-10 strains were homozygous. G222A-4-5 was homozygous, while the others were heterozygous. Nine lines of T239A-3 (3 strains), Y242A-4 (3 strains), and N422A-7 (3 strains), were heterozygous.

After PCR identification, RNA was extracted from the roots of three positive plants per T1 generation line. The transcript level of *CmPT2*-OE-8 was defined as 1. The expression levels of *CmPT2*-OE-9, -10, and -19 were 4.3, 4.96, and 3.1 times than those of *CmPT2*-OE-8, respectively. The expression levels of *CmPT2*-OE-1 and -11 were 2.48 and 2.25 times those of *CmPT2*-OE-8, respectively. The expression levels of *CmPT2*-OE-2 and -29 were 7% and 9% of *CmPT2*-OE-8, respectively (Appendix A). Expression of the post-translational modification-mutated *Cmphts* was tested in various lines of transgenic rice G84A, G222A, T239A, Y242A, and N422A. The RT-PCR results showed that among the 10 lines of G84A, the expression levels of *CmPT2*^G84A^ in G84A-10 and G84A-5 were 4.42 and 1.5 times those of *CmPT2* in *CmPT2*-OE-8. In nine lines of G222A, the expression levels of *CmPT2*^G222A^ were lower than those of *CmPT2* in *CmPT2*-OE-8. The expression levels of G222A-4 and -8 were 62% and 54% of *CmPT2*-OE-8, respectively. Among the 10 T239A lines, the highest relative expression of *CmPT2*^T239A^ was in T239A-3, approximately 1.16 times *CmPT2* expression in *CmPT2*-OE-8. The expression level of *CmPT2*^T239A^ in T239A-10 was 94% of that of *CmPT2* in *CmPT2*-OE-8. Among the nine lines of Y242A, the highest expression levels of *CmPT2*^Y242A^ were observed in Y242-A-4, approximately 1.23-fold higher than that of *CmPT2* in *CmPT2*-OE-8. Among the nine lines of N422A, the highest *CmPT2*^N422A^ expression was observed in N422A-7, approximately 2.17 times *CmPT2* expression in *CmPT2*-OE-8 (Appendix A).

### 2.4. Maturation Phenotype of T2 Generation in the CmPT2/Cmphts Overexpression Materials

The mature stage phenotype of *CmPT2*-OE T2 lines showed that the heights of *CmPT2*-OE-1-7, *CmPT2*-OE-4-2, *CmPT2*-OE-8-1, *CmPT2*-OE-10-1, and *CmPT2*-OE-11-5 were 1.04, 1.06, 1.02, and 1.07 times that of W7, respectively. However, the height of the *CmPT2*-OE-9-1 strain was slightly lower than that of the wild-type strain, ~92% of W7 (Figure 6A,C). *CmPT2*-OE-1-7, *CmPT2*-OE-8-1, *CmPT2*-OE-9-1, *CmPT2*-OE-10-1, and *CmPT2*-OE-11-5 had an increased number of productive panicles. *CmPT2*-OE-1-7 had 1.29 times more panicles than the wild-type. *CmPT2*-OE-4-2 had fewer panicles than W7 (approximately 77%) (Figure 6D). The panicle length (Figure 6B,E), primary branch number (Figure 6F), and secondary branch number (Figure 6G) of transgenic plants were all higher in *CmPT2*-OE than in W7. The yields of individual strains *CmPT2*-OE-1-7 and *CmPT2*-OE-10-1 were significantly higher than those of W7, which were 1.7-fold and 1.5-fold, respectively (Figure 6H).

We selected *Cmpht*-OE lines that had similar RNA expression levels as *CmPT2*-OE-8, as detected in the T1 generation. The mature stage phenotype of the T2 generation was detected under normal phosphate conditions. The data showed that the plant heights of G84A-10-3, G222A-4-1, T239A-3-1, Y242A-4-1, and N422A-7-1 were slightly lower than *CmPT2*-OE-8-1 (83%, 95%, 94%, 93%, and 88% of *CmPT2*-OE-8-1, respectively) (Figure 7A,C). Meanwhile, G222A-4-1 was about 1.1 times the number of productive panicles of *CmPT2*-OE-8-1; the number of productive panicles in other mutants was less than that in *CmPT2*-OE-8-1 (about 80–96%) (Figure 7D). The numbers of primary and secondary branches in all mutants except G222A-4-1 were lower than those in *CmPT2*-OE-8-1 (Figure 7F,G). The panicle length and yield among *Cmpht*-OE lines and *CmPT2*-OE-8 lines did not differ significantly (Figure 7B,E,H).

### 2.5. P/Pi Content of T2 Generation in the CmPT2/Cmphts Overexpression Plants

The effective phosphorus content was determined from rice leaves and roots after treatment under different P_i_ conditions. Under both HP and LP conditions, the phosphate content in leaves was not significantly different among *CmPT2*-OE, *Cmphts*-OE, and W7 (Figure 8A,C). However, under LP-stress conditions, *CmPT2*-OE strains had a higher phosphate content than wild-type roots (Figure 8B). The P_i_ content in the roots of *CmPT2*-OE-4-1 and *CmPT2*-OE-9-1 was 2.33 and 1.725 times that of W7 under LP stress, respectively (Figure 8C). Notably, the P_i_ content in the roots of N422A-7-1 was lower than that of *CmPT2*-OE-8-1 under LP conditions, although the P_i_ content of the other mutants was not significantly different from that of *CmPT2*-OE-8-1 (Figure 8D).

### 2.6. Expression Analysis of CmPT2/Cmphts-OE in T2 Generation Materials

The expression levels of phosphorus-related genes in transgenic and wild-type plants showed that the expression levels of *OsSPX2*, *OsIPS1*, *OsPT2*, *OsPT3*, *OsPT6*, *OsPT8*, and *OsPT10* were not significantly different between the W7 and *CmPT2*-OE plants under normal P_i_ conditions. Under LP conditions, the expression of these phosphorus starvation response (PSR) genes in transgenic plants was significantly lower than that in wild-type plants. This suggests that transgenic plants are less sensitive to LP stress than wild-type plants. The degree of PSR gene expression varied among different transgenic lines and was related to *CmPT2* expression levels. The PSR genes were expressed at low levels in *CmPT2*-OE-9-1, which had a high *CmPT2* expression level. However, they were highly expressed in *CmPT2*-OE-1-7 and *CmPT2*-OE-11-5, which showed low *CmPT2* expression levels in these lines (Figure 9).

The expression levels of the phosphorus-related genes in G84A-10-3, G222A-4-1, T239A-3-1, Y242A-4-1, and N422A-7-1 were similar to those in *CmPT2*-OE-8-1 under HP conditions. Under LP stress, *OsIPS1* and *OsSPX2* were increased in G84A-10-3, G222A-4-1, and T239A-3-1, similar to *CmPT2*-OE-8-1. By contrast, *OsIPS1* and *OsSPX2* increased weakly in Y242A-4-1 and N422A-7-1. The expression of *OsPT2* and *OsPT10* in T239A-3-1 was significantly higher than that in the other strains. Under LP, the expression levels of *OsPT3* and *OsPT6* were significantly higher in T239A-3-1, Y242A-4-1, and N422A-7-1 than in *CmPT2*-OE-8-1, G84A-10-3, and G222A-4-1. The expression of *OsPT8* was significantly higher, whereas that of *OsPT10* was significantly lower in *Y242A*-4-1 than in the other strains (Figure 10).

## 3. Discussion

### 3.1. CmPT2 Could Complete the Growth of Yeast Mutant MB192

Plants need to absorb phosphate at the root/soil interface through specific phosphate transporter proteins and complete the transmembrane transport of phosphorus between cells [6]. The heterologous transformation of genes is an important means of identifying gene function. There have been many reports on the functional identification of the *Pht* gene from different plant sources with functionally assisted validation by complementation of yeast phosphate transport-related mutants [14]. Analysis of the enzymatic kinetic activity of phosphate transport proteins can be performed using a ^32^P radioisotopic assay, which was used to verify the function of CmPT1 (AGK29560) [15]. The sequence identity between chrysanthemum gene *CmPT2* and CmPT1 is 90.8% [13]. In this study, *CmPT2* was able to complement the ability of the yeast phosphate transport deletion mutant MB192 to grow in LP environments, with a K_m_ of 65.7 μM using a ^32^P radioisotope. Yp112-*CmPT2* grew best at pH 4.0, indicating that CmPT2 belongs to the hydrogen proton co-transport high-affinity phosphate transporter protein and plays an important role in the phosphate absorption process in yeast. These results indicated that CmPT2 is a high-affinity phosphate transporter protein in chrysanthemums.

### 3.2. Five of the Nine Mutants of CmPT2 Could Not Complete the Growth of MB192

Furthermore, nine sites of protein post-translational modification (glycosylation, phosphorylation, myristoylation of CmPT2 protein) were analyzed. When serine ^514^Ser was mutated to aspartate (Asp), the plasma membrane localization changed to the endoplasmic reticulum (ER) [16]. However, in the subcellular localization results of transient expression in the onion epidermis, the extent of plasma membrane localization did not change after mutation in the post-translational modified amino acid sites (Appendix A). Studies in *Arabidopsis* have found that a mutation in Y in the AtPht1;1 protein disrupts the formation of its homodimer and enhances P_i_ uptake [17]. Here, complementation experiments with yeast mutant MB192 showed that G84A, G222A, T239A, Y242A, and N422A yeast transformants could not grow normally under LP conditions. The optimum pH of the five mutant yeast transformants was also changed. The greater the K_m_ value, the lower the affinity of the enzyme for the substrate and the lower the enzyme activity. The K_m_ values of these five mutants were much greater than that of Yp112-*CmPT2*, and we speculated that the affinity of the transporter for P_i_ was decreased. These data indicate that the loss of function or alteration in yeast was influenced by mutations in CmPT2 post-modified amino acid sites. We speculated that changes in the kinetic curve may be due to morphological changes in the protein underpinning its function.

### 3.3. CmPT2 May Improve Rice Yield by Increasing Phosphate Absorption

Heterologous transformation in gene function analysis has been reported in plants. Overexpression of the *Arabidopsis* high-affinity phosphate transporter gene PHT1 in tobacco culture cells can promote cell growth under LP conditions [18]. On expressing the barley high-affinity phosphate transporter gene Pht1;1 in rice, the phosphate absorption rate significantly increased [19]. Overexpression of the tobacco high-affinity phosphate transporter gene NtPT1 promoted phosphate uptake and accumulation in transgenic rice [20]. The *Brassica napus* phosphate transporter gene BnPht1;4 can promote phosphate resorption and improve the root architecture in *Arabidopsis* [21]. In the present study, statistics on the mature traits of T2 generation *CmPT2*-OE transgenic materials and wild-type plants showed that the growth status of *CmPT2*-OE materials was improved under field management conditions. The high P_i_ content in the roots of *CmPT2*-OE lines compared to that of W7 indicated that the overexpression of *CmPT2* may promote the absorption of phosphorus in rice and consequently improve yield in rice. Considering the RT-PCR results, we surmise that moderate expression levels of *CmPT2* may promote rice growth (*CmPT2*-OE-1-7). *CmPT2*-OE-9-1 lines had decreased plant height, yellow leaf tip, and reduced tiller number, whereas *CmPT2* was highly expressed and P_i_ content was the highest in *CmPT2*-OE-9-1. When the expression level of *CmPT2* was high, a certain degree of the phosphorus poisoning phenotype appeared. These results were similar to those of OsPht1; 8-overexpressing rice, which demonstrated excessive phosphorus accumulation and symptoms of phosphorus toxicity [7]. However, *CmPT2* was functionally redundant in a phosphate-adequate (300 µM) environment. Compared to W7, *CmPT2*-OE was less sensitive to LP stress and may be more adaptable to LP conditions.

### 3.4. Four of the Five Mutants Affected the Function of CmPT2 in Rice

To further confirm the effects of these five sites on CmPT2, we transformed them into the rice. The height of the T2-generation of G84A and N422A transgenic plants was lower than the *CmPT2*-OE-8-1-expressing plant, indicating that they affected the increase in growth by *CmPT2* in rice. The primary branch numbers of G84, T239A, and Y242A transgenic plants decreased, suggesting that they dropped from development by *CmPT2* in rice. PTMs play an important role in response to nutrient stress. Surprisingly, O-GlcNAcylation has extensive crosstalk with phosphorylation, where it serves as a nutrient/stress sensor to modulate signaling, transcription, and cytoskeletal functions [22]. The responses to P_i_ vs. phosphate (H_2_PO_3_^−^) vary in P_i_-starved *Arabidopsis* suspension cells, and the differences between them are mainly at the protein phosphorylation level [23]. Here, the effective phosphate content in the roots and leaves was lower in the N422A-7-1 strain than in the *CmPT2*-OE-8-1 strain, suggesting that the glycosylation of N422 of *CmPT2* may be important for the phosphate transport function of *CmPT2*. N422 is a glycosylation site located at TM2 and TM10. These results indicated that the transmembrane domains TM2 and TM10 play critical roles in the phosphate transport process of CmPT2 proteins. This may be related to the binding of H^+^ to P_i_ [9]. Myristoylation promotes protein association with the cell membrane and regulates membrane protein expression. However, this targeting mechanism during plant signal processing remains unknown [24]. In addition, T239A and Y242A, which serve as phosphorylation sites located on the intermediate hydrophilic ring, also play an important role in the phosphate transport process of CmPT2. Phosphorylation regulates the reception and transduction of cellular signals. It is a highly efficient signal for post-translational modifications of proteins in cells and acts in plants in response to abiotic and biotic stress conditions [25]. Therefore, four PTMs, G84, T239, Y242, and N422, might be the key amino acid sites in the function of CmPT2 in rice. However, G222A might have had little effect here. This suggests that these post-translational modification sites may be closely related to signal transduction at the cell membrane, perhaps via the recognition of H^+^ by P_i_ binding.

### 3.5. CmPT2 May Regulate the Expression of the PSR Genes Directly or Indirectly

PHR1 is considered a central regulator of the plant response to phosphorus deficiency stress, an intermediate vector from phosphorus signaling to downstream high-affinity phosphate transporters. It is involved in the expression of most downstream phosphorus-deficiency-responsive genes, including *SPX* (SPX domain protein) and *IPS* (induced by phosphate starvation) [26]. The SPX domain can indicate the phosphate state in fungal, plant, and human cells. In eukaryotes, inorganic phosphorus uptake, transport, storage, and signal transduction are inseparable from SPX domain-containing proteins [27]. In rice, a negative feedback regulatory pathway of phosphorus signaling and balance exists between *SPX2* and *PHR2* [28]. Rice *IPS1*, a member of the *TPSll*/*MT4* family, is involved in the regulation of the inorganic phosphate signaling pathway and homeostasis [29]. RT-PCR results showed that *SPX2* and *IPS1* were induced by phosphorus deficiency at the transcriptional level in wild-type rice. *SPX2* and *IPS1* were significantly inhibited, and the inhibition level correlated with the expression level of *CmPT2* in *CmPT2*-OE lines. These results suggest that *CmPT2* reduced the sensitivity of transgenic plants to LP stress. However, the expression patterns of P-related genes in the T239A-3-1, Y242A-4-1, and N422A-7-1 overexpression lines were different from those in the CmPT2-OE-8-1 strain.

The rice Pht1 family has 13 genes (*OsPT1*-*OsPT13*). OsPT2 is a low-affinity phosphate transporter. Overexpression of *OsPT2* caused a P-poisoning phenotype and a large increase in P content. Inhibition of *OsPT2* expression results in a low rate of shoot P uptake and a low total phosphorus content [30]. *OsPHT1;3* regulates the uptake and remobilization of P_i_ under extreme LP conditions [31]. OsPT6, a high-affinity phosphate transporter, can complement the yeast phosphate uptake mutant. Inhibition of *OsPT6* expression also reduces the rate of shoot P uptake and affects the ability to transport P to the shoots [32]. OsPT8, similar to OsPT6, fully complemented the yeast *pho84* mutants. Overexpression of OsPT8 causes a phosphotoxic phenotype in rice [7]. *OsPT9* and *OsPT10* are specifically induced by P starvation in the root epidermis, root hair, and lateral roots. Downregulation of *OsPT9* or *OsPT10* did not significantly reduce P uptake under low- or high-P conditions [33]. The function of OsPT9 or OsPT10 may be redundant for phosphate uptake. In this study, the expression of *Pht1* family genes was tested in wild-type and *CmPT2*-OE rice treated with different phosphate concentrations. Under LP conditions, the expression of phosphate transporter protein genes, *OsPT2*, *OsPT3*, *OsPT6*, *OsPT8*, and *OsPT10*, was strongly induced. The expression of *OsPT2*, *OsPT3*, *OsPT6*, and *OsPT8* was significantly lower in the *CmPT2*-OE lines than that in W7. The degree of PSR gene expression varied among transgenic lines and correlated with *CmPT2* expression levels. This shows that chrysanthemum phosphate transporter CmPT2 might replace the phosphate transporter in rice. However, the expression level of OsPT10 was not significantly different between *CmPT2*-OE and W7 plants, confirming the redundancy of *OsPT10*. However, under LP conditions, *OsPT3* and *OsPT6* expression levels were significantly higher in the T239A-3-1, Y242A-4-1, and N422A-7-1 lines than in the *CmPT2*-OE-8-1 strain. *OsPT8* expression was significantly increased, whereas *OsPT10* expression was significantly decreased in the Y242A-4-1 OE line. Thus, mutations in these three sites (two phosphorylation sites and one glycosylation site) may be important for the regulation of P-related genes by *CmPT2* directly or indirectly.

## 4. Methods and Materials

### 4.1. Yeast Strain and Plant Materials

The yeast mutant MB192 (*MATa* pho3-1 pho84::HIS3 ade2 leu2-3, 112 his3-532, trp1-289 ura3-1, 2 can1) was purchased from National BioResource Project (Yeast), Osaka University, Osaka Prefecture, Japan. The converted rice material was W7. *CmPT2* was cloned from chrysanthemums in our previous study [13].

### 4.2. Site-Directed Mutagenesis

Based on the restriction enzyme site in the yeast expression vector (p112A1NE), fusion high-fidelity PCR was performed to construct the yeast expression vector p112A1NE-*CmPT2* with 112-EcoR-F and 112-Not-R. After analysis of CmPT2 protein transmembrane domain and protein post-translational modification (glycosylation, phosphorylation, myristoylation) sites [9,13]. PCR-based site-directed mutagenesis was applied to the p112A1NE-*CmPT2* construct using fusion high-fidelity DNA polymerase based on primers designed according to the protocol of the Quickchange^®^ Site-Directed Mutagenesis Kit (Stratagene, La Jolla, CA, USA) [34]. All mutated primer sequences are listed in Appendix A. Primers were synthesized by Generay Biotech Co., Ltd. (Shanghai, China). The nine mutants were named G75A, G84A, G222A, T239A, Y242A, N422A, T425A, G456A, and S510A. All plasmid constructs were sequenced to ensure that no unexpected mutations or cloning errors occurred.

### 4.3. Construction of a Green Fluorescent Protein (GFP) Fusion Vector and Intracellular Localization Analysis

Site-directed mutagenesis was performed using the plasmid pENTR^TM^1A-*CmPT2* constructed in our previous studies [13]. All mutated constructs were recombined into the C-terminus GFP fusion vector pMDC43 by the LR reaction (as described in Gateway^®^ Technology with Clonase^®^ II) [35]. Plasmid DNA was bombarded into onion (*Allium cepa*) epidermal cells using a helium-driven gene gun (PDS-1000; Bio-Rad, Hercules; CA, USA) according to the manufacturer’s instructions [36]. The onion epidermal cells were incubated on Murashige and Skoog (MS) solid medium plates in the dark for 16–20 h. GFP expression was monitored using confocal laser scanning microscopy at 488 nm (Zeiss, Oberkochen, Germany) [37].

### 4.4. Functional Complementation Assay of CmPT2 and Cmphts in Yeast

The yeast expression vector p112A1NE carrying the p112A1NE empty vector and *CmPT2* and *Cmphts* open reading frame was transformed into MB192 (named Yp112, Yp112-*CmPT2*, and Yp112-*Cmphts*, respectively). The wild-type yeast strains, MB192, Yp112, Yp112-*CmPT2*, and Yp112-*Cmphts* were grown to the logarithmic phase and then spotted onto yeast nitrogen base medium plates containing different P_i_ concentrations (20, 60, and 100 mM) evenly [34]. The cells in the logarithmic phase were also transferred into the YNB medium containing 60 μM P_i_ for 40 h. The optical density of the yeast culture was measured every 8 h. To substantiate the pH dependence of P_i_ uptake, different extracellular pH values from 4 to 8 were used at a fixed amount of 80 mM KH_2_PO_4_. For ^32^P uptake experiments in yeast, approximately 5 mg of fresh yeast cell sample was used following a previously described method [38].

### 4.5. Regeneration of CmPT2 and Cmphts-Overexpressing Rice

The rice overexpression vectors pCAMBIA1300-*CmPT2* and pCAMBIA1300-*Cmphts* were constructed using p112A1NE-*CmPT2* and p112A1NE-*Cmphts* with the recombinant primers *CmPT2*-CE-F and *CmPT2*-CE-R, respectively. Primers were synthesized by Beijing Qingke Xinye Biotechnology Co., Ltd. (Beijing, China). The transgenic expression of *CmPT2/Cmphts* was carried out at Jian Gu Biotechnology Co., Ltd., Hefei, Anhui, China. Transgenic hygromycin-resistant plants overexpressing *CmPT2* or *Cmphts* were identified by PCR and RT-PCR at the DNA and RNA levels and named *CmPT2* or *Cmphts*-OE, respectively. Primer sequences are shown in Appendix A. The relative expression of *CmPT2* was calculated using the 2^−ΔΔCt^ method [39].

### 4.6. Treatments of Plant Materials

All experimental materials were planted in the experimental base of the China Rice Research Institute in Fuyang District, Hangzhou in summer (sown on 24 May, 2020 and transplanted on 20 June, 2020) and southern propagation base of the Hainan Lingshui Rice Research Institute in winter (sowed on 14 December, 2020 and transplanted on 14 January, 2021), with conventional field management. Fertilization occurred at the tillering and heading stages. The amount of N fertilizer was 165 kg/hm^2^, with m (base fertilizer):m (tillering fertilizer):m (panicle fertilizer) = 5:2:3. The amount of K fertilizer was 165 kg/hm^2^, with m (tiller fertilizer):m (ear fertilizer) = 7:3. The amount of P fertilizer was 90 kg/hm^2^ as base fertilizer. The phenotypes were counted under these conditions after ripening.

The rice materials were treated with nutrient solutions (Appendix A) of different phosphate concentrations. After seven days of hydroponics in a 96-cell rice hydroponic box, the seedlings were transferred to a plastic incubator filled with 20 L of different phosphate concentrations (300 μM P_i_, 10 μM P_i_) for 10 d [7] and subjected to RT-PCR and P_i_ content determination. Each rice seedling was fixed on a colonization plate with a colonization sponge. The nutrient solution was changed every three days during these periods. The expression of PSR genes and characteristics of T2 generations, W7, *CmPht1,2*-OE, G84A, G222A, T239A, Y242A, and N422A, were investigated. The soluble phosphate content was determined using the molybdenum blue method with three biological replicates [40].

### 4.7. Expression Analysis of Pi Absorption-Related Genes in CmPT2 and Cmphts-Overexpressing Rice

The template used for quantitative RT-PCR was the cDNA reverse recorded in the above step. The enzyme used was ChamQ Universal SYBR qPCR Master Mix (Novzan Biotechnology Co., Ltd., Nanjing, China). The relative expression of target genes was calculated using the 2^−ΔΔCt^ method [39]. Primers were synthesized by Qingke Xinye Biotechnology Co., Ltd. (Beijing, China) and are shown in Appendix A.

### 4.8. Statistical Analysis

All the significance discriminate analyses were carried out using the SPSS software (IBM SPSS Statistics Version 20), and the phenotypes of the OE lines and WT were compared by Student’s *t*-test at a 5% probability level.

## 5. Conclusions

This study explored the function of the chrysanthemum phosphate transporter CmPT2 in phosphate absorption in yeast and rice. CmPT2 belongs to the hydrogen proton co-transport high-affinity phosphate transporter. It plays an important role in the growth and phosphate uptake of yeast and rice. We also defined key post-translational modification amino acids affecting the function of CmPT2. The four PTM residues, G84, T239, Y242, and N422, might be the key amino acid sites for the function of CmPT2.

## Figures and Tables

**Figure 1 ijms-24-02025-f001:**
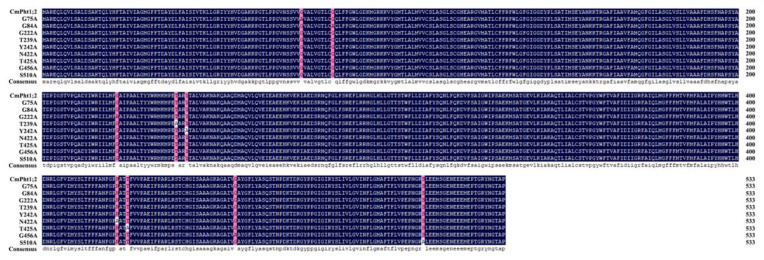
Sequences of 9 PTM mutation sites in CmPT2 phosphate transporter.

**Figure 2 ijms-24-02025-f002:**
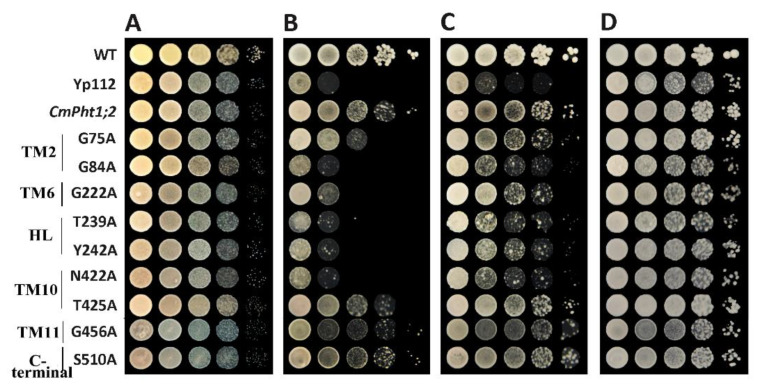
The growth of MB192 expression *CmPT2* and *Cmphts* under different Pi concentrations. (**A**) YPDA medium; (**B**–**D**) YNB medium supplemented with 20, 60 and 100 μM Pi, respectively. TM: the transmembrane domain, HL: hydrophilic ring, C-terminal: the hydrophilic chain of C terminal.

**Figure 3 ijms-24-02025-f003:**
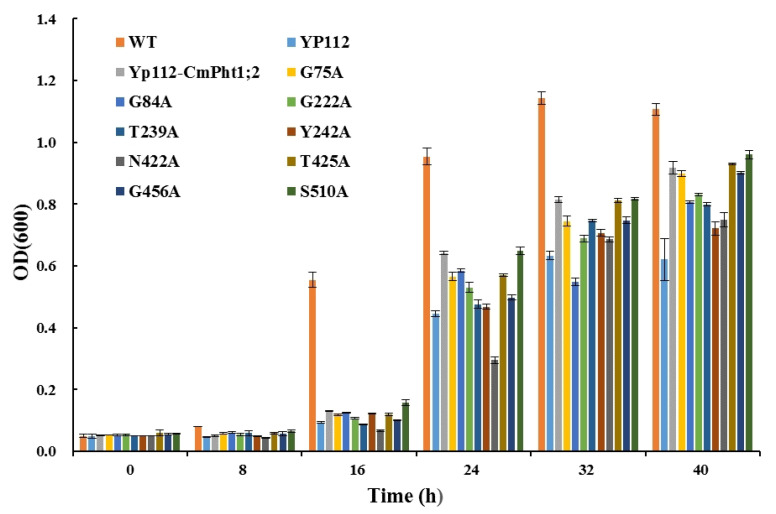
Growth rate of MB192 with *CmPT2* and *Cmphts* expression growing in YNB medium with 60 μM Pi.

**Figure 4 ijms-24-02025-f004:**
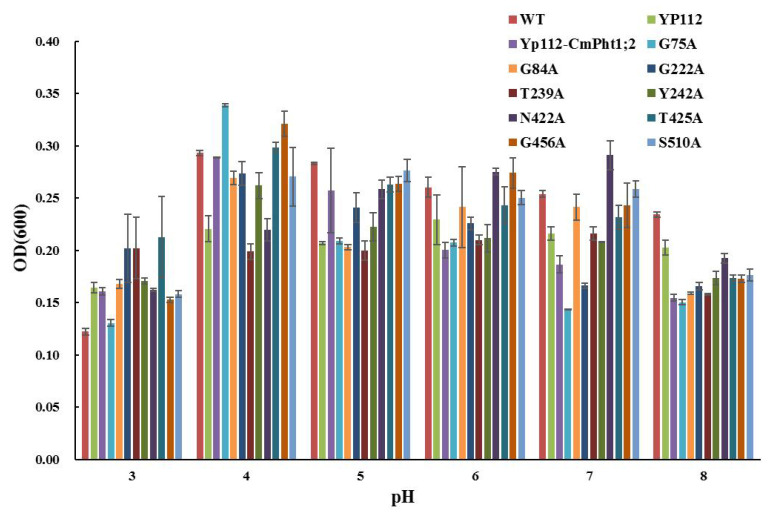
MB192 with *CmPT2* and *Cmphts* expression growing under different pH condition.

**Figure 5 ijms-24-02025-f005:**
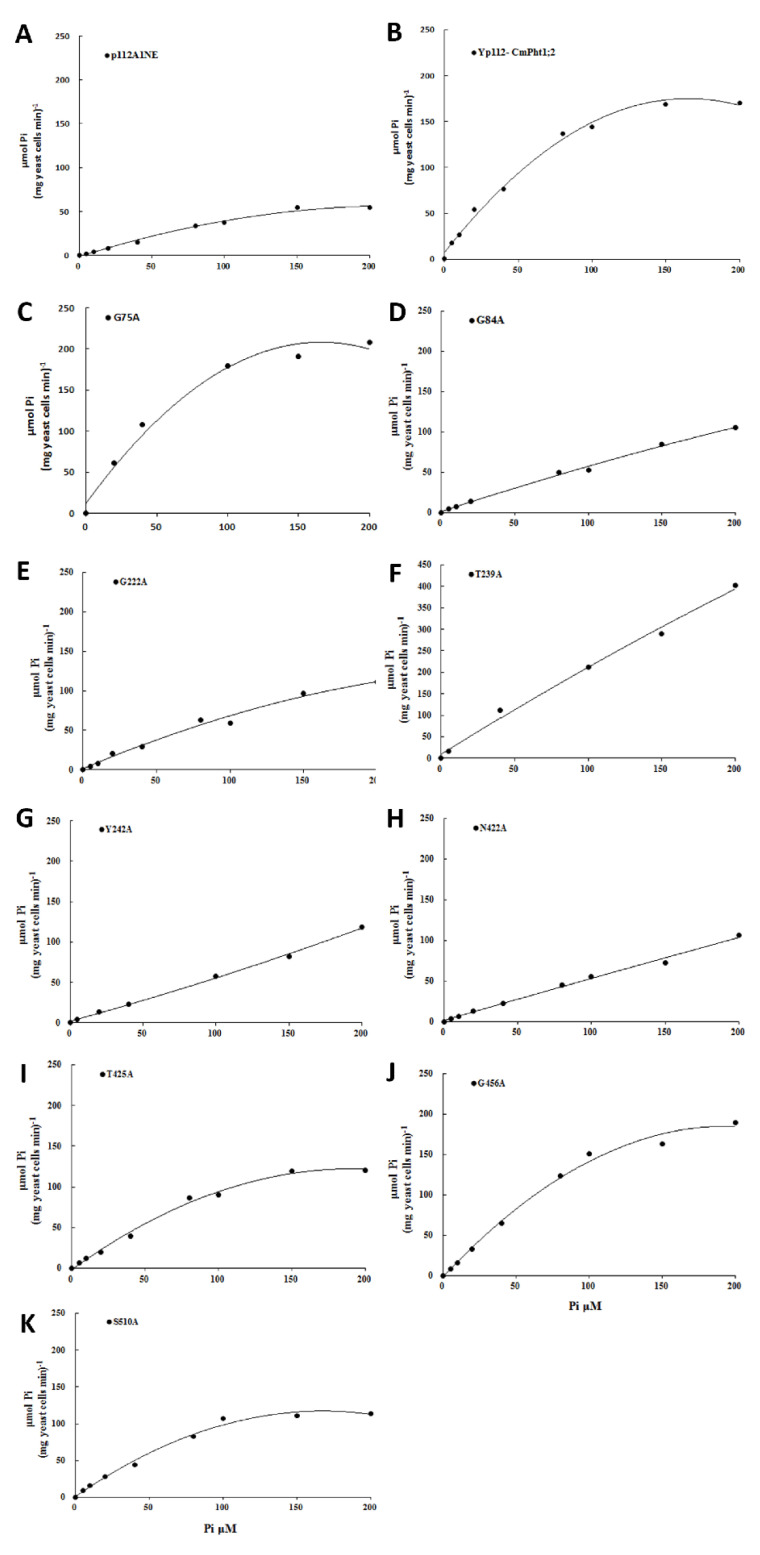
Velocity of ^32^Pi uptake of MB192 with *CmPT2* and *Cmphts* expression. (**A**): p112A1NE empty vector; (**B**): *CmPT2*; (**C**): G75A; (**D**): G84A; (**E**): G222A; (**F**): T239A; (**G**): Y242A; (**H**): N422A; (**I**): T425A; (**J**): G456A; (**K**): S510A.

**Figure 6 ijms-24-02025-f006:**
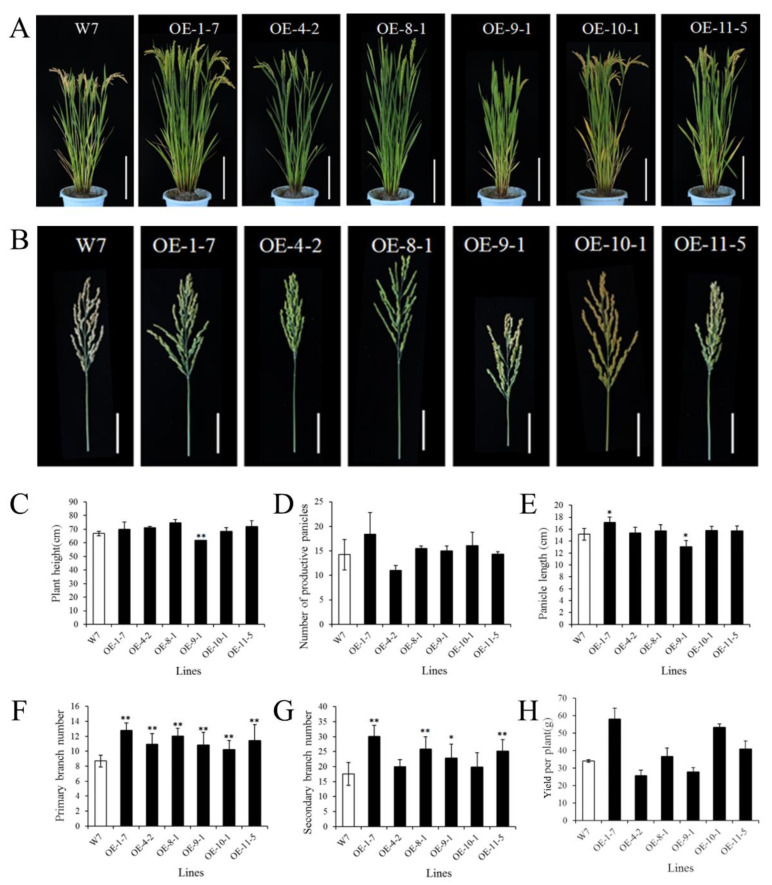
Phenotypes of *CmPT2*-OE T2 lines at the reproductive stage. (**A**): Plant architecture of wild type W7 and *CmPT2*-OE T2 lines at the reproductive stage. Bar = 20 cm. (**B**): Mature panicles of W7 and *CmPT2*-OE T2 lines. Bar = 5 cm. (**C**–**H**): Comparisons between W7 and *CmPT2*-OE T2 lines at the reproductive stage for average plant height (**C**), number of productive panicles (**D**), panicle length (**E**), number of primary branches (**F**), number of secondary branches (**G**), and yield per plant (**H**). Values are given as the mean ± SD. “**” and “*” indicates a significant difference at the 0.01 and 0.05 level, respectively.

**Figure 7 ijms-24-02025-f007:**
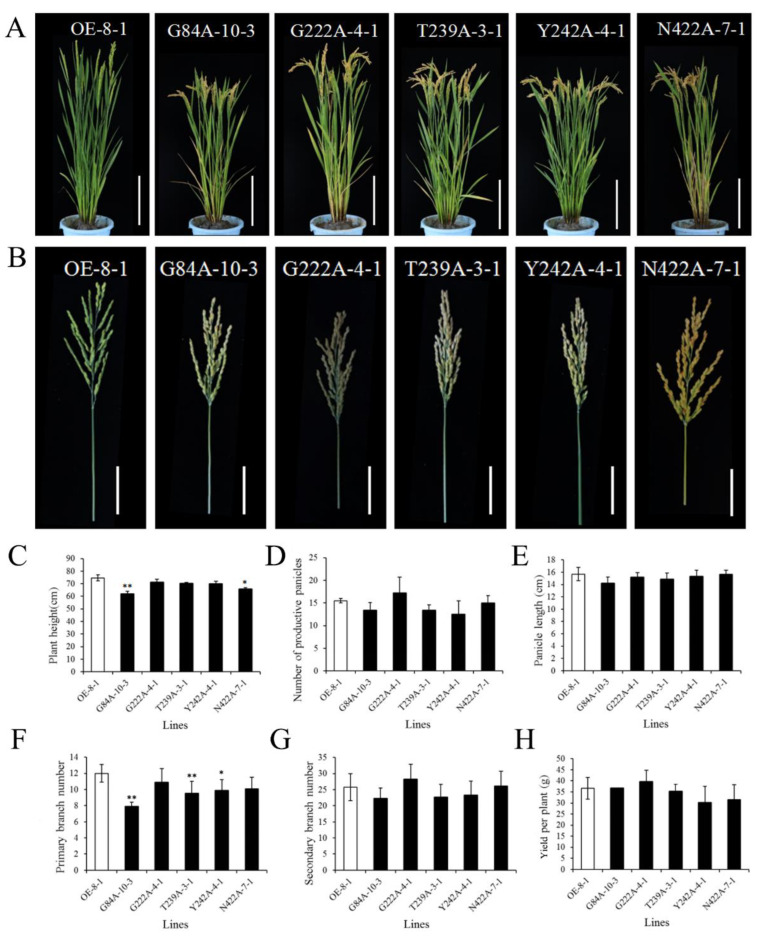
Phenotypes of *Cmphts*-OE T2 lines at the reproductive stage. (**A**): Plant architecture of *CmPT2*-OE-8-1 and *Cmphts*-OE lines at the reproductive stage. Bar = 20 cm. (**B**): Mature panicles of *CmPT2*-OE-8-1 and *Cmphts*-OE lines. Bar = 5 cm. (**C**–**H**): Comparisons between *CmPT2*-OE-8-1 and *Cmphts*-OE lines at the reproductive stage for average plant height (**C**), number of productive panicles (**D**), panicle length (**E**), number of primary branches (**F**), number of secondary branches (**G**), and yield per plant (**H**). Values are given as the mean ± SD. “**” and “*” indicates a significant difference at the 0.01 and 0.05 level, respectively.

**Figure 8 ijms-24-02025-f008:**
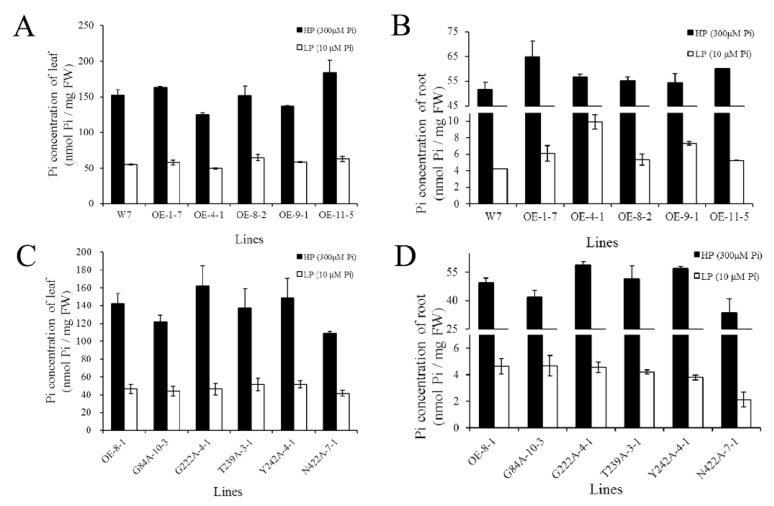
The Pi concentration of *CmPT2*-OE T2 lines under different phosphorus conditions. (**A**,**C**): Pi concentration of leaf; (**B**,**D**): Pi concentration of root; A, B: Pi concentration of W7 and *CmPT2*-OE lines; (**C**,**D**): Pi concentration of *CmPT2*-OE-8-1 and *Cmphts*-OE lines.

**Figure 9 ijms-24-02025-f009:**
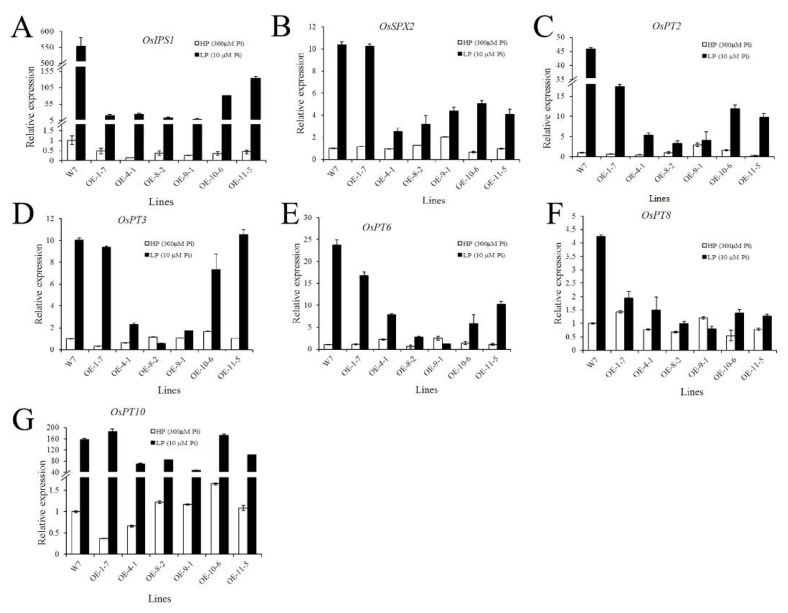
The relative expression analysis of *CmPT2* and other PSR gene under different phosphate conditions. Relative expression analysis of *OsIPS1* (**A**), *OsSPX2* (**B**), *OsPT2* (**C**), *OsPT3* (**D**), *OsPT6* (**E**), *OsPT8* (**F**), and *OsPT10* (**G**) between W7 and *CmPT2*-OE lines under different phosphate conditions.

**Figure 10 ijms-24-02025-f010:**
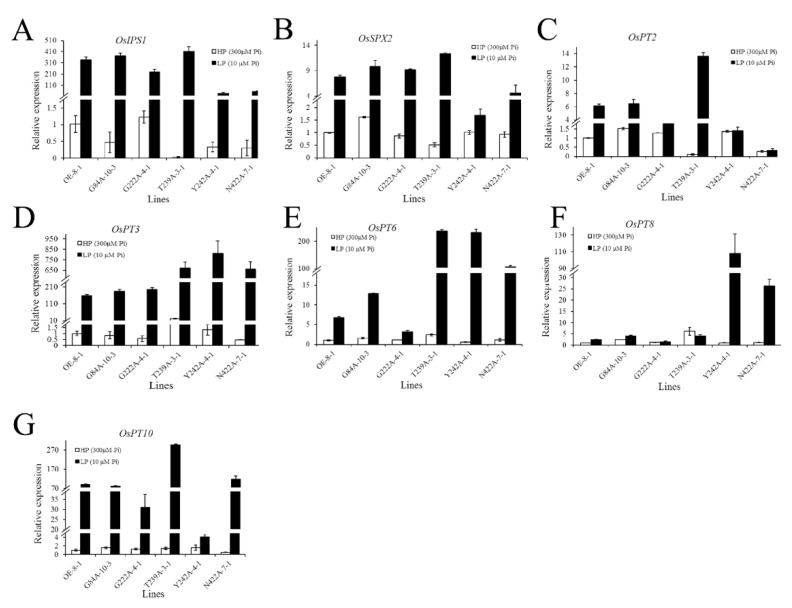
The relative expression analysis of *Cmphts* and other PSR gene under different phosphate conditions. Relative expression analysis of *OsIPS1* (**A**), *OsSPX2* (**B**), *OsPT2* (**C**), *OsPT3* (**D**), *OsPT6* (**E**), *OsPT8* (**F**), and *OsPT10* (**G**) between *CmPT2*-OE and *Cmphts-OE* lines under different phosphate conditions.

## Data Availability

Data is contained within the article or Appendix A.

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
