# Peer review of "Five Post-Translational Modification Residues of CmPT2 Play Key Roles in Yeast and Rice"

_ijms, 2023, doi:10.3390/ijms24032025_

Round 1
Reviewer 1 Report
Comment 1: Change the title
Comment 2: Write the abbreviation for CmPht1:2 protein.
Comment 3: In the abstract, details about the topic i.e introduction is not clear, please include at least two sentences before objectives.
Comment 4: Rewrite the abstract, clearly.
Comment 5: Please include line numbers.
Comment 6: Lot of information in the results section up to 2.7. But discussion only three topics. Please improve the discussion section.
Comment 7: Started abstract with Chrysanthemum but not even a single piece of information about the species in the introduction section, started with Phosphorus, in the discussion started with rice. Please keep the flow from the title to the conclusion.
Author Response
Dear reviewers and editors,
Thank you very much for your constructive opinion. I am very sorry for the delay in submitting the revisions. I had already response to all the questions in the attached files so far as I know. Please check and notice me if there is any questions. Thanks again.
Best wishes to you.
Sincerely yours,
Chen Liu
2023.01.09
Response to Reviewer 1
Comment 1: Change the title
Response: Thank you very much for your instructive comments. We have changed the title into: “Five Post-translational Modifications Residues of CmPht1;2 Play Key Roles in Yeast and Rice”(page 1, line 2-3).
Comment 2: Write the abbreviation for CmPht1;2 protein.
Response: CmPht1;2 protein (CmPT2). We have changed “CmPht1;2” into “CmPT2” in the whole article.
Comment 3: In the abstract, details about the topic i.e introduction is not clear, please include at least two sentences before objectives.
Response: The details had been added in the abstract. “Phosphate transporter Pht1 family member CmPht1;2 protein (CmPT2) plays an important role in response to low-phosphate (LP) stress in chrysanthemum. Post-translational modification (PTM) can modulate the function of proteins in multiple ways. Here, we used yeast and rice systems to study the role of putative PTM in CmPT2 by determining the effect of mutation of key amino acid residues of putative glycosylation, phosphorylation, and myristoylation sites.” (page 1, line 11-16).
Comment 4: Rewrite the abstract, clearly.
Response: Thank you very much for your instructive comments. We had rewritten the abstract. (page 1, line 10-34).
Comment 5: Please include line numbers.
Response: The line numbers has been added.
Comment 6: Lot of information in the results section up to 2.7. But discussion only three topics. Please improve the discussion section.
Response: Thank you very much for your instructive comments. We had improved the discussion section, especially the sections of 3.1-3.4 (page 12-13, line 250-338).
Comment 7: Started abstract with Chrysanthemum but not even a single piece of information about the species in the introduction section, started with Phosphorus, in the discussion started with rice. Please keep the flow from the title to the conclusion.
Response: Thank you very much for your instructive comments. We had reorganized the logical relationship of the whole manuscript (page 2, line 68-72; page 12-13, line 250-338).

Reviewer 2 Report
This manuscript by Tang et al. deals with CmPht1;2, a phosphate transporter from the flowering plant Chrysanthemum (Chrysanthemum morifolium Ramat.). Authors found that CmPht1;2 processes nine putative post-translational modifications (PTMs) sites that possibly play a crucial role under phosphate (Pi) limited conditions. To test that hypothesis, independent CmPht1;2 mutants (Cmpht) were generated by abolishing the PTM site. Subcellular localization of Cmphts showed the plasma membrane similar to CmPht1;2. Furthermore, the yeast complementation assay suggested that CmPht1;2 is involved in Pi absorption and could recover mutant growth in the low Pi condition.
After that, they overexpress the CmPht1;2 and the five Cmpht mutants into rice cultivar ‘Wuyunjing 7’ (W7), and plant phenotype was evaluated. They found that plant height, adequate panicle numbers, branch numbers, and yield of CmPht1; 2-overexpression (OE) lines were higher than wild-type. On the other hand, Cmphts-OE plants showed smaller plant height and panicle numbers than CmPht1;2-OE. Several phosphorus-starvation response gene expressions were downregulated in low Pi conditions in CmPht1;2-OE lines. Overall, the authors claimed that they characterized the key PTM sites of CmPht1;2 that determine the protein function under Pi.
Firstly, this manuscript needs extensive English language editing.
Writing comments for this is a tough job as line numbers need to be included. However, I am going to add some comments and corrections.
Sub-cellular localization of any protein is determined by the localization signal-peptide commonly present at the C or N-terminus of any protein. I believe a point mutation in the protein can not alter the localization unless the mutation occurs in the signal peptide. The mislocalization of the protein reported by (Bayle et al., 2011) is possibly the result of the disruption of the signal sequence. Hence, this experiment is divergent from the focus of the manuscript. However, they can discuss that and provide such data as supplementary.
This is recommended to provide more detailed “field management” procedures and methods.
I was wondering about the root phenotype under low Pi conditions.
PTMs are dynamic events during the life cycle including nutrient stress like low Pi. Authors can pinpoint any outcome from this dataset related to phosphate homeostasis.
Fig. 9 No significant differences in root Pi were observed among WT, CmPht1;2, and CmPht1;2-OE T2 lines under 300uM phosphorus conditions. What is the significance of this result?
“Protein translational modifications (PTMs) increase the functional diversity of the proteins through covalent addition of func-tional groups or proteins, proteolytic cleavage of regulating subunits, or degradation of the entire protein (Uversky, 2013).”- Post-translational modifications…
“The effect of these potential modified sites of CmPht1;2 on the phosphate transport function needs to be further verified. In present study, we created 9 mutants of these PTMs (Cmphts) trough point mutation technology.” point mutation technology? It’s just a tool.
“cauliflower mosaicvirus” or cauliflower mosaic virus.
“DNA was extracted from the T1 and T2 generations of hygromycin resistance plants by SDS and the transgenic plants were identified with” what is SDS??
I also suggest adding the prospects for this work regarding crop improvement and application.
Author Response
Dear reviewers and editors,
Thank you very much for your constructive opinion. I am very sorry for the delay in submitting the revisions. I had already response to all the questions in the attached files so far as I know. Please check and notice me if there is any questions. Thanks again.
Best wishes to you.
Sincerely yours,
Chen Liu
2023.01.09
Response to Reviewer 2
This manuscript by Tang et al. deals with CmPht1;2, a phosphate transporter from the flowering plant Chrysanthemum (Chrysanthemum morifolium Ramat.). Authors found that CmPht1;2 processes nine putative post-translational modifications (PTMs) sites that possibly play a crucial role under phosphate (Pi) limited conditions. To test that hypothesis, independent CmPht1;2 mutants (Cmpht) were generated by abolishing the PTM site. Subcellular localization of Cmphts showed the plasma membrane similar to CmPht1;2. Furthermore, the yeast complementation assay suggested that CmPht1;2 is involved in Pi absorption and could recover mutant growth in the low Pi condition.
After that, they overexpress the CmPht1;2 and the five Cmpht mutants into rice cultivar ‘Wuyunjing 7’ (W7), and plant phenotype was evaluated. They found that plant height, adequate panicle numbers, branch numbers, and yield of CmPht1; 2-overexpression (OE) lines were higher than wild-type. On the other hand, Cmphts-OE plants showed smaller plant height and panicle numbers than CmPht1;2-OE. Several phosphorus-starvation response gene expressions were downregulated in low Pi conditions in CmPht1;2-OE lines. Overall, the authors claimed that they characterized the key PTM sites of CmPht1;2 that determine the protein function under Pi.
Q1: Firstly, this manuscript needs extensive English language editing.
Response: Thank you very much for your instructive comments. We have made extensive English language editing.
Q2: Writing comments for this is a tough job as line numbers need to be included. However, I am going to add some comments and corrections.
Response: The line numbers has been added.
Q3: Sub-cellular localization of any protein is determined by the localization signal-peptide commonly present at the C or N-terminus of any protein. I believe a point mutation in the protein can not alter the localization unless the mutation occurs in the signal peptide. The mislocalization of the protein reported by (Bayle et al., 2011) is possibly the result of the disruption of the signal sequence. Hence, this experiment is divergent from the focus of the manuscript. However, they can discuss that and provide such data as supplementary.
Response: Thank you very much for your instructive comments. We had transferred the relative results (Figure 6) into Supplementary Figure S2, and deleted relative script of Results section. We had already discussed that and provide such data as supplementary (page 7, line 133; page 12, line 267-272). “When serine 514Ser was mutated to aspartate (Asp), the plasma membrane localization changed to the endoplasmic reticulum (ER) (Bayle et al., 2011). However, in the subcellular localization results of transient expression in the onion epidermis, the extent of plasma membrane localization did not change after mutation in the post-translational modified amino acid sites (Supplementary Figure S2).”
Q4: This is recommended to provide more detailed “field management” procedures and methods.
Response: Thank you very much for your instructive comments. We had added more details about “field management” in Methods and Materials (page 15-16, line 445-453). “All experimental materials were planted in the experimental base of the China Rice Research Institute in Fuyang District, Hangzhou in summer (sown on May 24 and transplanted on June 20) and southern propagation base of the Hainan Lingshui Rice Research Institute in winter (sowed on December 14 and transplanted on January 14), with conventional field management. Fertilization occurred at the tillering and heading stages. The amount of N fertilizer was 165 kg/hm2, with m (base fertilizer):m (tillering fertilizer):m (panicle fertilizer) = 5:2:3. The amount of K fertilizer was 165 kg/hm2, with m (tiller fertilizer):m (ear fertilizer) = 7:3. The amount of P fertilizer was 90 kg/hm2 as base fertilizer. The phenotypes were counted under these conditions after ripening.
”
Q5: I was wondering about the root phenotype under low Pi conditions.
Response: The phenotype was tested at the field conditions and the roots were hardly harvested.
Q6: PTMs are dynamic events during the life cycle including nutrient stress like low Pi. Authors can pinpoint any outcome from this dataset related to phosphate homeostasis.
Response: Thank you very much for your instructive comments. We added about the events of PTMs involved in nutrient stress in the discussion (page 13, line 315-320). “PTMs play an important role in response to nutrient stress. Surprisingly, O-GlcNAcylation has extensive crosstalk with phosphorylation, where it serves as a nutrient/stress sensor to modulate signaling, transcription, and cytoskeletal functions (Hart et al., 2011). The responses to Pi vs. phosphate (H2PO3-) vary in Pi-starved Arabidopsis suspension cells, and the differences between them are mainly at the protein phosphorylation level (Mehta et al., 2021)..”
Q7: Fig. 9 No significant differences in root Pi were observed among WT, CmPht1;2, and CmPht1;2-OE T2 lines under 300uM phosphorus conditions. What is the significance of this result?
Response: This result might indicate that expression level of CmPht1;2 in WT is enough for Pi up-taking under phosphate-adequate (300 uM) environment (Page 13, line 305-306). “However, CmPT2 was functionally redundant in a phosphate-adequate (300 µM) environment. ”
Q8: “Protein translational modifications (PTMs) increase the functional diversity of the proteins through covalent addition of func-tional groups or proteins, proteolytic cleavage of regulating subunits, or degradation of the entire protein (Uversky, 2013).”- Post-translational modifications…
Response: We have changed “Protein translational modifications” into “Post-translational modifications” (Page 2, line 54).
Q9: “The effect of these potential modified sites of CmPht1;2 on the phosphate transport function needs to be further verified. In present study, we created 9 mutants of these PTMs (Cmphts) trough point mutation technology.” point mutation technology? It’s just a tool.
Response: We have changed “technology” into “method” (Page 2, line 77-78). “In present study, we created 9 mutants of these PTMs (Cmphts) trough point mutation method.”
Q10: “cauliflower mosaicvirus” or cauliflower mosaic virus.
Response: The results of subcellular location of Cmphts were deleted (page 7, line 133).
Q11: “DNA was extracted from the T1 and T2 generations of hygromycin resistance plants by SDS and the transgenic plants were identified with” what is SDS??
Response: We have added the full name of “SDS”. “DNA was extracted from the T1 and T2 generations of hygromycin-resistant plants using sodium dodecyl sulfate (SDS)...” (Page 7, line 136-137).
Q12: I also suggest adding the prospects for this work regarding crop improvement and application.
Response: Thank you very much for your instructive comments. We have added the prospects in the abstract. “This work bolsters our understanding of the function of phosphate transporters and provides genetic resources for improving the efficiency of phosphorus utilization in crop plants.” (Page 1, line 32-34).

Round 2
Reviewer 1 Report
Comment 1: Please check author instructions about citation of references in the main text. Is it author name or number?
Comment 2: If possible write the conclusion.
Author Response
Dear reviewers and editors,
Thank you very much for your constructive opinion. we had made a revision of the citation of references in the main text and written the conclusion. Please check and notice me if there is any questions. Thanks again.
Best wishes to you.
Sincerely yours,
Chen Liu
2023.01.11
Response to Reviewer 1
Comment 1: Please check author instructions about citation of references in the main text. Is it author name or number?
Response: Thank you very much for your instructive comments. The citation of references in the main text should be number and we had made a revision.
Comment 2: If possible write the conclusion.
Response: Thank you very much for your instructive comments. We had written the conclusion “5. Conclusion. This study explored the function of the chrysanthemum phosphate transporter CmPT2 in phosphate absorption in yeast and rice. CmPT2 belongs to the high-affinity phosphate transporter. It plays an important role in the growth and phosphate uptake of yeast and rice. We also defined key post-translational modification amino acids affecting the function of CmPT2. The four PTM residues, G84, T239, Y242, and N422, might be the key amino acid sites in the function of CmPT2.”(page 12, line 461-467).

Reviewer 2 Report
The authors provided satisfactory comments.
Author Response
Dear reviewers and editors,
Thank you very much for your approvement. Please check and notice me if there is any questions. Thanks again.
Best wishes to you.
Sincerely yours,
Chen Liu
2023.01.11